# Lower Limbs Wearable Sports Garments for Muscle Recovery: An Umbrella Review

**DOI:** 10.3390/healthcare10081552

**Published:** 2022-08-16

**Authors:** João P. Duarte, Ricardo J. Fernandes, Gonçalo Silva, Filipa Sousa, Leandro Machado, João R. Pereira, João P. Vilas-Boas

**Affiliations:** 1Porto Biomechanics Laboratory (LABIOMEP-UP), University of Porto, 4200-450 Porto, Portugal; 2Research Unity in Sport and Physical Activity (CIDAF, UID/DTP/04213/2020), Faculty of Sport Sciences and Physical Education, University of Coimbra, 3040-248 Coimbra, Portugal; 3Faculty of Sport (CIFI2D), University of Porto, 4099-002 Porto, Portugal; 4Faculty of Physical Education and Sport, Lusofona University of Humanities and Technology, 1749-024 Lisboa, Portugal

**Keywords:** wearable textile, recovery, fatigue, exercise, injury, heating, muscle damage (DOMS)

## Abstract

This review aims to understand the different technologies incorporated into lower limbs wearable smart garments and their impact on post-exercise recovery. Electronic searches were conducted in the PubMed, Web of Science, and Cochrane electronic databases. Eligibility criteria considered meta-analyses that examined the effects of wearable smart garments on physical fitness in healthy male and female adults. Seven meta-analyses were considered in the current umbrella review, indicating small effects on delayed-onset muscle soreness ([0.40–0.43]), rate of perceived exertion (0.20), proprioception (0.49), anaerobic performance (0.27), and sprints ([0.21–0.37]). The included meta-analyses also indicated wearable smart garments have trivial to large effects on muscle strength and power ([0.14–1.63]), creatine kinase ([0.02–0.44]), lactate dehydrogenase (0.52), muscle swelling (0.73), lactate (0.98) and aerobic pathway (0.24), and endurance (0.37), aerobic performance (0.60), and running performance ([0.06–6.10]). Wearing wearable smart garments did not alter the rate of perceived exertion and had a small effect on delayed-onset muscle soreness. Well-fitting wearable smart garments improve comfort and kinesthesia and proprioception and allow a reduction in strength loss and muscle damage after training and power performance following resistance training or eccentric exercise.

## 1. Introduction

Categorized into three significant areas (clothing, electronics, and information science) [1], many wearable electrogarment systems have emerged on the market in the last decade [2]. These textile-based systems measure biological signals (such as body temperature, electroencephalogram, electrocardiogram, electro-oculography, surface electromyogram, galvanic skin response, and respiration) and can be used for detecting and monitoring medical conditions, and for supporting post-exercise recovery and rehabilitation [3]. These garments can intervene in different areas, particularly for monitoring general consumers’ daily physical exercise, and for screening physical conditions, performance, and recovery.

Different techniques, such as cold-water immersion, massage, and dynamic recovery procedures, might positively affect post-exercise recuperation although their effectiveness remains ambiguous [4]. Complementarily, novel interventions (such as compression garments and ice vests) need more evidence-based data to support their applicability and success. Subjective recovery markers, assessed using well-being questionnaires, have been shown to have high reproducibility [5] and can be used concurrently with more traditional physiological indicators (such as blood lactate concentration, creatine kinase, lactate dehydrogenase, and aspartate-aminotransferase enzymes) [6]. Meanwhile, heart rate variability and muscle activation are arising as attractive alternatives to delineate the physical conditioning status and the readiness for more precise performances [4].

Previous research has focused on wearable garment technology applications [7,8,9,10] but focused on the medical or healthcare areas, giving less priority to post-exercise recovery. Furthermore, the available variety of smart garment applications specifically considering the lower limbs is very limited. Since umbrella reviews provide a ready information summary, simplifying evidence-based planning and decision-making [11], we aimed to better systematize and understand the different array of technologies incorporated into lower limbs wearable smart garments and their impact on post-exercise recovery (based on physiological and perceived exertion outcomes).

## 2. Materials and Methods

The current umbrella review was conducted following previous recommendations [11] and addressed all items suggested in the PRISMA statement [12]. The study protocol was registered with PROSPERO (CRD42021238799).

### 2.1. Literature Search

A computerized systematic literature search was performed in the PubMed, Web of Science, and Cochrane Library databases. A Boolean search syntax was used (Table 1) and was limited to full-text availability, publication before 31 December 2021, adult subjects, English language, and type of article (meta-analysis). An additional search within the included studies’ reference lists was conducted to retrieve additional relevant meta-analyses to be included in the current umbrella review.

### 2.2. Selection Criteria

Based on a priori defined inclusion/exclusion criteria (population, intervention, comparator, outcome, and study design-PICOS; Table 1), two independent reviewers (JPD and GS) screened potentially relevant articles by analyzing their titles, abstracts, and full texts to clarify their eligibility. When JPD and GS did not reach agreement concerning an article inclusion, a third independent reviewer (JRP) was compelled to decide. The descriptive analyses focused on different outcome categories (delayed-onset muscle soreness, muscle strength, creatine kinase, blood lactate concentration, lactate dehydrogenase, muscle swelling, muscle power, proprioception, sprints, maximum oxygen uptake, rate of perceived exertion, and aerobic and anaerobic performances). The information regarding the literature search, selection criteria, and considered moderator variables is presented in Table 1.

### 2.3. Methodological Quality Evaluation

The identification of meta-analyses of different sources of bias in randomized controlled trials is critical to distinguish between low and high quality. For this purpose, each included meta-analysis was independently assessed by three reviewers (JPD, JRP, and GS; Table 2) using the A Measurement Tool to Assess Systematic Reviews (AMSTAR2) [13]. This checklist contains 16 literature search procedures, data extraction, quality assessment, and statistical analyses, with each item being fulfilled with a yes, partial yes, or no (1, 0.5, and 0 points, respectively). The high-, moderate-, and low-quality result corresponded to ≥80, 40–79, and <40% of the possible score [14].

### 2.4. Quality Evidence

Using the modified Grading of Recommendations Assessment, Development and Evaluation (GRADE) principles [15], for every single outcome of the included meta-analyses the following were analyzed: (i) the study limitations (using the risk of bias scales in the primary studies of the included meta-analyses); (ii) the inconsistency (through the statistical heterogeneity size, i.e., I2-statistics); (iii) the indirectness (by evaluating differences between study cohorts, intervention types, comparators, and outcome variables of the primary studies and those relevant for each included meta-analysis); (iv) the imprecision (using the 95% confidence interval width of the included meta-analyses’ pooled effect size); and (v) the publication bias (examining the included meta-analyses’ funnel plots asymmetry). Each one of the above-referred points was evaluated for every single outcome as not reported, neutral, serious, or very serious [15]. Firstly, meta-analyses were downgraded from four points by one point for each not reported or serious and by two points for each very serious rating. Then, they were rated as high-, moderate-, low-, or very-low-quality evidence (4, 3, 2, and <1 points, respectively). The GRADE assessment (Table 3) was conducted independently by three researchers (JPD, JP, and GS) that discussed and agreed on any differences.

### 2.5. Prediction Interval

The 95% prediction interval, standardized mean difference, upper limits of the 95% confidence interval, and tau-squared values were calculated for all included meta-analyses. These values were obtained according to the Comprehensive meta-analysis v3 software [16] and the previous literature [14].

### 2.6. Data Interpretation

The magnitude of effects across all included meta-analyses was compared (Table 4) and the standardized mean difference values were classified as <0.20 trivial, 0.20–0.50 small, 0.50–0.80 medium, and ≥0.80 large effects [17].

**Table 2 healthcare-10-01552-t002:** General characteristics of the included systematic review and meta-analyses studies.

Study	Design	Age (mean ± SD)	Included Studies	Sample Size	Garment Recovery Method	Outcome	AMSTAR Quality
Brown et al. (2017) [18]	Meta-analysis	25.0 ± 9.0	23	348 (256 M/92 F)	Compression	Muscle strength and power, endurance, and sprints	Moderate
Ghai et al. (2016) [19]	Meta-analysis	28.0 ± 15.0	50	1443 (627 M/719 F)	Joint stabilizers Compression	Proprioception	Moderate
Hill et al. (2014) [20]	Meta-analysis	22.3 ± 2.3	12	205 (136 M/69 F)	Compression	Delayed-onset muscle soreness, muscle strength, and creatine kinase	Moderate
Marques-Jimenez et al. (2016) [21]	Meta-analysis	23.6 ± 3.0	20	279 (169 M/99 F/11 NR)	Compression	Blood lactate concentration, creatine kinase, lactate dehydrogenase, muscle swelling, strength and power, and delayed-onset muscle soreness	Moderate
da Silva et al. (2018) [22]	Meta-analysis	29.5 ± 5.9	23	294 (249 M/45 F)	Compression	Running time, maximal oxygen uptake, and rate of perceived exertion	High
Douzi et al. (2019) [23]	Meta-analysis	NR	45	473	Cooling Ice vests	Aerobic and anaerobic performances	Moderate
Altarriba-Bartes et al. (2020) [24]	Meta-analysis	20.8 ± 1.3	5	69 M	Compression	Counter movement jump, 20 m sprint, and maximal voluntary contraction	Moderate

Abbreviations: Standard deviation (SD), not reported (NR), males (M), and females (F).

**Table 3 healthcare-10-01552-t003:** Quality of evidence for each outcome of the included meta-analyses using Grading of Recommendations Assessment, Development and Evaluation (GRADE).

Meta-Analysis	Outcome	GRADE Items					Quality of the Evidence
		Risk of Bias	Inconsistency	Indirectness	Imprecision	Publication Bias	
Brown et al. (2017) [18]	Muscle strength	Serious	Serious	No serious	No serious	Not reported	Very low
	Muscle power		Serious	No serious	No serious		
	Endurance		Serious	No serious	No serious		
	Sprints		Serious	No serious	No serious		
Ghai et al. (2016) [19]	Proprioception	No serious	No serious	No serious	No serious	Likely	Moderate
Hill et al. (2014) [20]	Delayed onset of muscle soreness	No blinding	No serious	No serious	No serious	Not reported	Low
	Muscle strength		No serious	No serious	No serious		
	Creatine kinase		No serious	No serious	No serious		
Marques-Jimenez et al. (2016) [21]	Blood lactate concentration	Serious	Serious	No serious	No serious	Not reported	Very low
	Creatine kinase	Serious	Serious	No serious	No serious		
	Lactate dehydrogenase	Serious	Serious	No serious	No serious		
	Muscle swelling	Serious	Serious	No serious	No serious		
	Muscle strength	Serious	Serious	No serious	No serious		
	Muscle power	Serious	Serious	No serious	No serious		
	Delayed onset of muscle soreness	Serious	Serious	No serious	No serious		
da Silva et al. (2018) [22]	Running performance	No blinding	Serious	No serious	No serious	Undetected	Moderate
	Maximal oxygen uptake		Serious	No serious	No serious		
	Rate of perceived exertion		Serious	No serious	No serious		
Douzi et al. (2019) [23]	Aerobic performance	Serious	No serious	No serious	No serious	Likely	Moderate
	Anaerobic performance		No serious	No serious	No serious		
Altarriba-Bartes et al. (2020) [24]	Counter movement jump	Serious (−1)	No serious	No serious	No serious	Undetected	Moderate
	20 m sprint		No serious	No serious	No serious		
	Maximal voluntary contraction		Serious (−1)	No serious	No serious		

**Table 4 healthcare-10-01552-t004:** Included meta-analyses that examined the effects of smart compression garments on physiological outcomes in healthy adults.

Meta-Analysis	Outcome	Effect Size/Mean Difference (95% CI, *p* Value); I^2^ (Chi^2^, *p* Value)	Prediction Interval
Brown et al. (2017) [18]	Muscle strength	Mean difference: 0.37 (0.22–0.51, n.a.); 66% (n.a., *p* ≤ 0.001)	0.37 (−1.12–1.86)
	Muscle power		
	Endurance		
	Sprints		
Ghai et al. (2016) [19]	Proprioception	Hedge’s g: 0.49 (0.36–0.62, *p* ≤ 0.001); 24% (n.a., *p* = 0.26)	0.49 (−1.54–2.52)
Hill et al. (2014) [20]	Delayed-onset muscle soreness	Hedge’s g: 0.40 (0.24–0.57, *p* ≤ 0.001); 0.001% (n.a.)	0.40 (−1.16–1.96)
	Muscle strength	Hedge’s g: 0.46 (0.22–0.70, *p* ≤ 0.001); 4.8% (n.a.)	0.46 (−1.37–2.29)
	Muscle power	Hedge’s g: 0.49 (0.27–0.71, *p* ≤ 0.001); 0.001% (n.a.)	0.49 (−1.32–2.30)
	Creatine kinase	Hedge’s g: 0.44 (0.17–0.70, *p* ≤ 0.001); 37.4% (n.a.)	0.44 (−1.36–2.24)
Marques-Jimenez et al. (2016) [21]	Blood lactate concentration	Mean difference: 0.98 (0.28–1.68, n.a.); 80% (60.48, *p* ≤ 0.001)	0.98 (−1.98–3.94)
	Creatine kinase	Mean difference: −0.02 (−0.44–0.40, n.a.); 83% (166.24, *p* ≤ 0.001)	0.02 (−1.37–1.41)
	Lactate dehydrogenase	Mean difference: −0.52 (−1.42–0.38, n.a.); 81% (26.83, *p* ≤ 0.001)	0.52 (−2.72–3.76)
	Muscle swelling	Mean difference: −0.73 (−1.20–−0.26, n.a.); 75% (75.58, *p* ≤ 0.001)	0.73 (−1.04–2.50)
	Muscle strength	Mean difference: 1.18 (0.84–1.51, n.a.); 78% (196.08, *p* ≤ 0.001)	1.18 (−1.36–3.72)
	Muscle power	Mean difference: 1.63 (1.10–2.16, n.a.); 85% (195.84, *p* ≤ 0.001)	1.63 (−1.38–4.64)
	Delayed-onset muscle soreness	Mean difference: −0.43 (−0.66–−0.19, n.a.); 68% (148.60, *p* ≤ 0.001)	0.43 (−0.27–1.13)
da Silva et al. (2018) [22]	Running performance 50–400 m	Mean difference: 0.06 (1.99–2.11, n.a.); 0% (n.a., *p* = 0.922)	0.06 (−5.52–5.64)
	Running performance 800–3000 m	Mean difference: 6.10 (−7.23–19.43, n.a.); 0% (n.a., *p* = 0.991)	6.10 (−12.23–24.43)
	Running performance >5000 m	Mean difference: 1.01 (−84.80–86.82, n.a.); 0% (n.a., *p* = 0.999)	1.01 (−123.27–125.00)
	Maximal oxygen uptake	Mean difference: 0.24 (−1.48–1.95, n.a.); 0% (n.a., *p* = 1.000)	0.24 (−3.39–3.87)
	Rate of perceived exertion	Mean difference: −0.20 (−0.48–0.08, n.a.); 0% (n.a., *p* = 0.982)	0.20 (−0.76–1.16)
Douzi et al. (2019) [23]	Aerobic performance	Mean difference: 0.60 (0.43–0.77, n.a.); 36% (n.a., *p* ≤ 0.001)	0.60 (−1.49–2.69)
	Anaerobic performance	Mean difference: 0.27 (0.04–0.50, n.a.); 31% (n.a., *p* < 0.05)	0.27 (−1.42–1.96)
Altarriba-Bartes et al. (2020) [24]	Counter movement jump 24 h	Mean difference: 0.14 (−0.31–0.59, n.a.); 0% (n.a., *p* = 0.59)	0.14 (−10.05–10.32)
	Counter movement jump 48 h	Mean difference: 0.69 (0.14–1.25, n.a.); 27% (n.a., *p* = 0.26)	0.69 (−13.96–15.34)
	20 m sprint 24 h	Mean difference: −0.28 (−0.81–0.24, n.a.); 0% (n.a., *p* = 0.75)	n.c.
	20 m sprint 48 h	Mean difference: −0.21 (−0.74–0.31, n.a.); 0% (n.a., *p* = 0.52)	n.c.
	Maximal voluntary contraction 24 h	Mean difference: 0.57 (−1.10–2.25, n.a.); 88% (n.a., *p* ≤ 0.001)	n.c.
	Maximal voluntary contraction 48 h	Mean difference: 0.23 (−0.39–0.84, n.a.); 0% (n.a., *p* = 0.99)	n.c.

Abbreviations: CI (confidence interval); n.a. (not applicable); n.c. (not computable).

## 3. Results

### 3.1. Search Results

A total of 122 potentially relevant studies were identified in the electronic databases (Figure 1) and 7 meta-analyses were eligible for inclusion in the current umbrella review based on the a priori selection criteria.

### 3.2. Meta-Analyses Characteristics

The included meta-analyses were published between 2013 and 2020, the number of included original studies ranged from 5–50 (33 on average), and the sample sizes were between 69 and 1443 trained and untrained healthy adults (>18 years old). Five meta-analyses investigated the effects of compression garments [18,19,20,21,22], one meta-analysis was centered on joint stabilizers [19], and another focused on cooling ice vests [23]. The methodological quality evaluation (AMSTAR2) of the included meta-analyses is summarized in Table 2. The included papers were classified from 44–80% of the maximum score (16 points), with six [18,19,20,21,23,24] and one [22] meta-analyses being of moderate and high methodological quality, respectively. The included meta-analyses’ quality of evidence (GRADE) assessment is summarized in Table 3. Three of the included studies [18,20,21] presented evidence of very low and low quality, and four studies [19,22,23,24] provided evidence of moderate quality.

### 3.3. Effectiveness of Lower Limbs Wearable Sports Garments 

The encompassed meta-analyses indicated small effects on the subjective delayed-onset muscle soreness ([0.40–0.43]), rate of perceived exertion (0.20), and proprioception (0.49) variables [19,20,22], and on the anaerobic pathway, particularly anaerobic performance (0.27) and sprints ([0.21–0.37]) [18,23,24]. The included meta-analyses also indicated trivial to large effects of wearable smart garments on muscle strength and power ([0.14–1.63]) [18,20,21,24]; the physiological variables creatine kinase ([0.02–0.44]), lactate dehydrogenase (0.52), muscle swelling (0.73), and blood lactate concentration (0.98) [20,21]; and on the aerobic pathway, namely maximum oxygen uptake (0.24), endurance (0.37), aerobic performance (0.60), and running performance ([0.06–6.10]) [18,22,23] in healthy male and female adults (Table 4).

## 4. Discussion

The current systematic umbrella review aimed to provide an overview of the effects of lower limbs wearable smart garments on post-exercise recovery (using physiological and perceived exertion outcomes) in healthy male and female adults. The main finding is that the lower limbs wearable smart garments have small effects on subjective variables (particularly on delayed-onset muscle soreness, rate of perceived exertion, and proprioception) and on the anaerobic pathway (through sprinting ability), and trivial to large effects on muscle strength and power, physiological variables (creatine kinase, lactate dehydrogenase, muscle swelling, and blood lactate concentration), and the aerobic pathway (maximum oxygen uptake and running performance). Complementarily, we observed that the included meta-analyses are of moderate to high methodological quality.

The meta-analyses of our umbrella review indicate a trivial effect of wearable smart garments on rate of perceived exertion in line with previous studies in athletes [25] and non-athletes [26]. In addition, we observed that wearing smart garments with optimized compression, fitting, and skin contact characteristics has a small effect on proprioception. Wearable smart garments that were well-fitting, comfortable, and kinesthesia improved single lower limb stance with closed eyes in healthy active females [27] and drop punt kick accuracy in elite football players [28]. Both studies evidenced that the group skill influences proprioception, with the poor inherent proprioceptive feedback cluster performing better with the application of wearable smart garments than their high-skilled counterparts. Likewise, wearable smart garments have a small effect on delayed-onset muscle fatigue, which is beneficial for athletes and may improve an individual’s readiness to participate in physical activity [29]. Although the mechanism explaining the cause of delayed-onset muscle fatigue currently remains unclear [20], the use of wearable smart garments generates an external pressure gradient that influences the osmotic pressure and reduces the space available for muscle swelling and hematoma to occur [30].

In the seven considered meta-analyses, the measurement of muscular strength focused on the assessment of isometric, isokinetic, or isotonic contractions with a dynamometer. Even if previous meta-analyses showed small effects of wearable smart garments on muscle strength [18,20,24], their effect on 2–8- and 24-h recovery is evident. Subsequently, eight studies focused on the effect of wearable smart garments on post-exercise muscle strength, including participants with different experience levels to non-strength-trained men and active or endurance-trained women [21]. The effects of wearable smart garments indicate faster recovery of muscle function after exercise (standard mean difference = 1.18). It is well demonstrated that the most significant effects of wearable smart garments on strength recovery appear at 3–8 (2.33–2.98) [31], 24 (1.01), 48 (1.47), 72 (1.57), and 96 h (1.88) [21], in agreement with the previous literature that identified their potential to reduce strength loss after a fatiguing exercise. Furthermore, the use of wearable smart garments during exercise can decrease sport-related musculoskeletal injury risk [32].

In the current study of meta-analyses, muscular power assessment focused on the evaluation of explosive power using squat and counter-movement jumps, resistance exercises at various loads and velocities, and a 5-m sprint bout. Furthermore, wearable smart garments’ elasticity and compression during exercise, aiming to enhance power production, do not elicit any improvement in maximal power [33]. These authors also highlighted that wearable smart garments’ positive impacts on muscle damage along explosive exercises would vary according to the outcome measures. This is described in the current umbrella review, with meta-analyses indicating small to large effects of wearable smart garments on muscle power, with two [18,20] and one [21] evidencing small and large effects, respectively. However, the clarification could be due to the different number of studies examined in the meta-analysis (30 vs. 96) [18]. Moreover, the different movements’ recovery rate and uniqueness of the neuromuscular profile were suggested previously [34].

The included meta-analyses indicated trivial to large effects of wearable smart garments on creatine kinase, lactate dehydrogenase, muscle swelling, and blood lactate concentration [20,21], with the literature not supporting their effect on the recovery of physiological and inflammatory variables [35,36,37]. It is known that compression, massage, and electrostimulation from wearable smart garments reduces the space available for swelling and inflammation to occur [30], and that the pressure from these dispositives may promote venous return, allowing for the removal of metabolic waste products [38]. Either way, while the use of wearable smart garments during exercise is still unclear, their effectiveness in supporting post-exercise recovery is evident and well-established [18,20,21].

The AMSTAR2 was developed to evaluate systematic reviews of randomized trials but not to generate a quality overall score. Nevertheless, with further steps to base more decisions on real-world observational evidence, this tool should help to identify high-quality systematic reviews [13]. In the current umbrella review, only one study registered the protocol [24], appropriate methods for statistical combination of results was not performed, and none reported the original studies’ funding sources. This might be due to word, table, and figure restrictions; the databases lack of supplemental materials [13]; and, eventually, to the fact that authors were unaware of the importance of these methodological quality characteristics.

The included meta-analyses presented very low, low, or moderate (two, one, and four studies, respectively) quality of evidence, possibly due to under-reported GRADE items that also downgraded the quality of evidence [14]. The following criteria were not sufficiently addressed in the analyzed meta-analyses: (i) #2, establish methods before conducting the meta-analyses; (ii) #11, use appropriated methods for statistical combination of results; (iii) #12, assess the risk of bias and potential impact in individual studies; and (iv) #15, carry out an adequate investigation of publication bias and discuss its likely impact on the review results.

The current umbrella review presents findings on the highest level of the evidence pyramid regarding wearable smart garments’ effects on physical fitness in healthy adult males and females. Furthermore, it ensured a high-level synthesis of potentially moderating variables and addressed the methodological quality and the quality of evidence. Finally, this umbrella review identified current gaps in the literature, allowing the proposal of suggestions for future research. A limitation of the current review is the (very) low evidence of some of the included meta-analyses and the fact that some of the assessed AMSTAR2 and GRADE criteria were under-reported or under-represented.

## 5. Conclusions

Wearing wearable smart garments during exercise did not alter the rate of perceived exertion and had a small effect on delayed-onset muscle soreness. Wearable smart garments that were well-fitting, comfortable, and kinesthesia improved proprioception and reduced strength loss and muscle damage after training and power performance following resistance training or eccentric exercise. Additionally, the American College of Sports Medicine (ACSM) in the 2022 worldwide survey of fitness trends [39], considering thousands of professionals worldwide, indicated wearable technology as the number one trend (including fitness or activity trackers, garments, smartwatches, heart rate monitors, and Global Positioning System (GPS) tracking devices). These devices can be used, for instance, as a step counter and to track heart rate, body temperature, spent calories, sitting time, and sleep time and quality, with innovations including blood pressure, oxygen saturation, body temperature, respiratory rate, electromyography, and electrocardiogram [39]. Research with high methodological quality and a high level of evidence should be conducted in the future.

## Figures and Tables

**Figure 1 healthcare-10-01552-f001:**
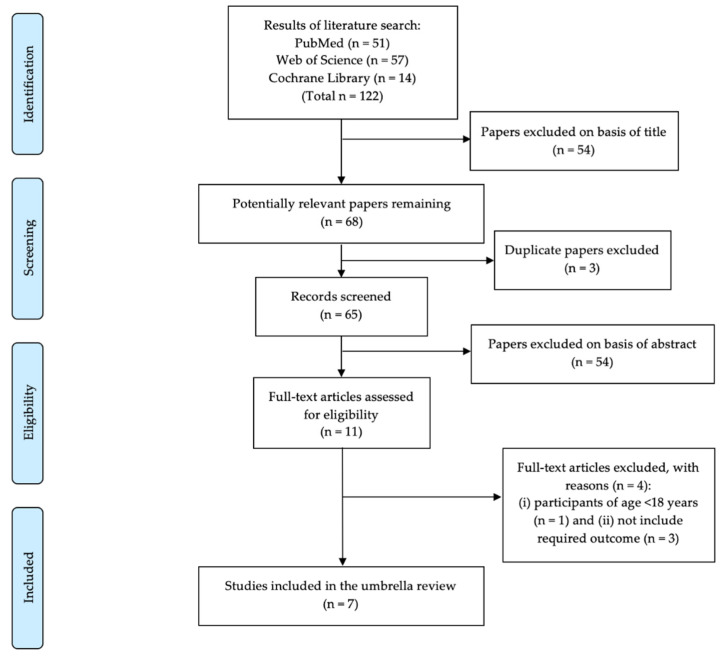
PRISMA flow chart representing the study screening and selection process.

**Table 1 healthcare-10-01552-t001:** Information on the literature search, selection criteria, and considered moderator variables.

Literature Search	Search Syntax	(garment OR tight OR stocking OR garments OR tights OR stockings) AND (compression OR recovery OR heat OR electrostimulation OR massage) AND (exercise OR EIMD OR performance OR recovery OR sport OR athlete) AND (meta-analysis)
Selection criteria	Population	Healthy adults (mean age > 18 years)
	Intervention	Lower limbs garments using different associated recovery methods (e.g., compression, massage, electrostimulation, or heat)
	Comparator	Control groups or groups that have been subject to different recovery protocols
	Outcome	At least one measure of muscle strength, muscle power, linear sprint speed, sprint/speed/agility, blood lactate concentration, creatine kinase, rate of perceived exertion, and delayed-onset muscle soreness
	Study design	Meta-analysis
Potential moderator variables	Chronological age Sex Expertise level	Adults Males and females Trained and untrained individuals

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
