# Peer review of "Lower Limbs Wearable Sports Garments for Muscle Recovery: An Umbrella Review"

_healthcare, 2022, doi:10.3390/healthcare10081552_

Round 1
Reviewer 1 Report
LOWER LIMBS WEARABLE SPORTS GARMENTS FOR MUSCLE RECOVERY: AN UMBRELLA REVIEW
This paper presents a review aiming to explore the different technologies incorporated into lower limbs wearable smart garments and their impact on post-exercise recovery. The paper electronic search was conducted in PubMed, Web of Science and Cochrane electronic databases. Eligibility criteria consider meta-analyses that examined the effects of wearable smart garments on physical fitness for males and females. Seven meta-analyses were considered in the current umbrella review indicating various effect sized and the general conclusions were that wearing wearable smart garments did not alter the rate of perceived exertion and had a small effect on muscle soreness delayed onset. In addition, the wearable smart garment's well-fitting, comfort and kinesthesia improve proprioception.
This paper was easy to read and to review since it is well written and well a perfumed meta-analytic work, systematically followed the protocols e.g. PRISMA, and it presents the findings in an organized way. It is hard to comment and suggest any revision, because the protocol is so detailed and correctly followed by the authors.
The finding are clearly presented and interpreted, while the implications are disused within the current interest of the research field.
In my opinion, the paper meets the standard for an international journal, and I endorse publication in this form.
Author Response
Thank you for your consideration of the protocol and outcomes organization.
The group of authors appreciated your comment on data presentation, interpretation, and discussion. Also, we prized your final recommendation.
Reviewer 2 Report
In this review paper, electronic searches were conducted in PubMed, Web of Science and Cochrane electronic databases. Eligibility criteria consider meta-analyses that examined the effects of wearable smart garments on physical fitness in healthy male and female adults. Seven meta-analyses were considered in the current umbrella review indicating small effects on the delayed-onset muscle soreness ([0.40-0.43]), rate of perceived exertion 20 (0.20), proprioception (0.49), anaerobic performance (0.27) and sprints ([0.21-0.37)]. The included meta-analyses also indicated wearable smart garments trivial-to-large effects on muscle strength and power ([0.14-1.63]), creatine kinase ([0.02-0.44]), lactate dehydrogenase (0.52), muscle swelling (0.73), lactate (0.98) and aerobic pathway (0.24), as well as on endurance (0.37), aerobic performance (0.60) and running performance ([0.06-6.10]). The results show that wearing wearable smart garments did not alter the rate of perceived exertion and had a small effect on muscle soreness delayed onset. The wearable smart garments well-fitting, comfort and kinesthesia improve proprioception and allow reducing strength loss and muscle damage after training and power performance following resistance training or eccentric exercise.
The paper is clear and well-organized. The Reviewer believes it is a good complement to this field and will help to understand the different technologies incorporated into lower limbs wearable smart garments. I suggests to accept in present form.
Author Response
The group of authors valued your comment on data presentation and interpretation, organization, and conclusions. Also, we appreciate your final recommendation.
Reviewer 3 Report
Dear Authors,
Manuscript: LOWER LIMBS WEARABLE SPORTS GARMENTS FOR 2 MUSCLE RECOVERY: AN UMBRELLA REVIEW
There is a major issue in the flow of the paper; the paper in the current format looks like a summary of different studies rather than providing the states of arts and highlighting the advantages and limitations of each approach as well as the future direction and application level. Lots of numbers and measures there but no clarification about what it means; how it can be improved; what kind of limitations we have, challenges…etc.
In line 67, the Authors mentioned that the last review for this topic is performed on 31/12/2021 and looking at the references they have only one paper published in 2022 so is this worth another review?
What are the additional relevant meta-analyses included in the current umbrella review?
In the first statement of the introduction, there is something missing maybe a word or a statement (read the statement carefully).
Author Response
There is a major issue in the flow of the paper; the paper in the current format looks like a summary of different studies rather than providing the states of arts and highlighting the advantages and limitations of each approach as well as the future direction and application level. Lots of numbers and measures there but no clarification about what it means; how it can be improved; what kind of limitations we have, challenges…etc.
A: Thank you for your considerations. Our umbrella review provides a ready information summary, simplifying evidence-based studies. For example, previous research has focused on wearable garment technology applications in the medical area, giving less priority to post-exercise recovery. The current study aimed to systematize better and understand the different technologies incorporated into lower limbs wearable smart garments and their impact on post-exercise recovery (based on physiological and perceived exertion outcomes).
In line 67, the Authors mentioned that the last review for this topic is performed on 31/12/2021 and looking at the references they have only one paper published in 2022 so is this worth another review?
A: Thank you for your remark. The meta-analysis included in this study were published until 2021. However, the included reference of 2022 concerned the relevance of this topic (wearable technology), which was considered by the American College of Sports Medicine on the worldwide survey of fitness trends, indicating wearable technology as the number one trend. Therefore, this reference of 2022 did not present any object or study type to be included in the umbrella review.
What are the additional relevant meta-analyses included in the current umbrella review?
A: Meta-analyses conclusions are described in table 4.
In the first statement of the introduction, there is something missing maybe a word or a statement (read the statement carefully).
A: Appreciated. The phrase was reformulated.
Round 2
Reviewer 3 Report
Dear Authors,
Many thanks for your hardworking to improve the manuscript.
This manuscript is a resubmission of an earlier submission. The following is a list of the peer review reports and author responses from that submission.